# Functional Role of Piezo1 in the Human Eosinophil Cell Line AML14.3D10: Implications for the Immune and Sensory Nervous Systems

**DOI:** 10.3390/biom14091157

**Published:** 2024-09-14

**Authors:** Sung-Min Hwang, Ji-Min Song, Jung Ju Choi, YunJae Jung, Chul-Kyu Park, Yong Ho Kim

**Affiliations:** 1Gachon Pain Center and Department of Physiology, College of Medicine, Gachon University, Incheon 21999, Republic of Korea; unclehwang76@gmail.com (S.-M.H.); a06025@gachon.ac.kr (J.-M.S.); 2Department of Health Science and Technology, Gachon Advanced Institute for Health Science & Technology, Gachon University, Incheon 21999, Republic of Korea; yjjung@gachon.ac.kr; 3Department of Anesthesiology and Pain Medicine, Gachon University, Gil Medical Center, Incheon 21565, Republic of Korea; jjchoi2@gilhospital.com; 4Department of Microbiology, College of Medicine, Gachon University, Incheon 21999, Republic of Korea

**Keywords:** eosinophil cell line, Piezo, dorsal root ganglion, TRPV1, purinergic receptors

## Abstract

Mechanosensitive ion channels, particularly Piezo channels, are widely expressed in various tissues. However, their role in immune cells remains underexplored. Therefore, this study aimed to investigate the functional role of Piezo1 in the human eosinophil cell line AML14.3D10. We detected *Piezo1* mRNA expression, but not *Piezo2* expression, in these cells, confirming the presence of the Piezo1 protein. Activation of Piezo1 with Yoda1, its specific agonist, resulted in a significant calcium influx, which was inhibited by the Piezo1-specific inhibitor Dooku1, as well as other nonspecific inhibitors (Ruthenium Red, Gd^3+^, and GsMTx-4). Further analysis revealed that Piezo1 activation modulated the expression and secretion of both pro-inflammatory and anti-inflammatory cytokines in AML14.3D10 cells. Notably, supernatants from Piezo1-activated AML14.3D10 cells enhanced capsaicin and ATP-induced calcium responses in the dorsal root ganglion neurons of mice. These findings elucidate the physiological role of Piezo1 in AML14.3D10 cells and suggest that factors secreted by these cells can modulate the activity of transient receptor potential 1 (TRPV1) and purinergic receptors, which are associated with pain and itch signaling. The results of this study significantly advance our understanding of the function of Piezo1 channels in the immune and sensory nervous systems.

## 1. Introduction

Eosinophils are multifunctional leukocytes that play pivotal roles in inflammatory processes [1]. Originating from bone marrow hematopoietic stem cells, eosinophils feature cytoplasmic granules containing cytotoxic proteins, cytokines, chemokines, and lipid mediators [2]. These cells migrate from the circulatory system to the site of inflammation, modulating immune responses to diverse stimuli [3]. The recruitment and activation of eosinophils at inflammatory sites are highly regulated processes, involving an intricate interplay among various receptors, ion channels, and signaling pathways [4,5,6]. Over recent decades, studies have identified novel roles of eosinophils across various pathological processes associated with eosinophilic diseases, including allergic rhinitis, asthma, and atopic dermatitis [7]. These conditions are characterized by a significant increase in the number of eosinophils within the blood and tissues, accompanied by the release of numerous and diverse cytokines, growth factors, and chemokines [8]. Conversely, eosinophils synthesize cytokines, chemokines, and growth factors that influence immune responses and may participate in the host defense mechanism against parasitic, viral, fungal, and bacterial infections [9]. In addition, recent research has revealed that eosinophils exhibit mechanosensitive properties, which enable them to detect fluid shear stress, resulting in their activation [10].

The activation of many stretch-activated or mechanically gated ion channels, which facilitate the passage of ions such as calcium in response to increased membrane tension, influences immune cell functions [11,12]. For example, transient receptor potential (TRP) channels, such as TRP vanilloid-type 4 (TRPV4), are broadly recognized as pivotal contributors to macrophage activation that influences the immune response to inflammatory agonists [13]. Recently, studies have identified the crucial contribution of piezo channels to mechanotransduction, with preferential Ca^2+^ influx being involved in various physiological functions such as cell volume homeostasis, somatosensation, vascular function, and proprioception [14,15]. One of the members of these channels, Piezo1, is highly sensitive to membrane tension and is activated in response to pressure and blood flow-induced shear stress [16,17], whereas Piezo2 is primarily expressed in sensory neurons and acts as a major mechanotransducer in proprioception [18,19,20]. Piezo1 is strongly expressed in immune cells and plays a vital role in inflammatory responses [21,22,23]. However, previous studies have not elucidated the role of Piezo1 in eosinophils.

Primary sensory neurons in the dorsal root ganglia (DRG) possess a variety of receptors associated with the sensory perception of pathological conditions such as chronic pain and itch in the peripheral nervous system [24,25]. In these pathological conditions, the immune system reportedly regulates neuronal function in the DRG via the secretion of proinflammatory factors [26,27,28]. Thus, we hypothesized that cytokines released from the human eosinophil cell line AML14.3D10 could regulate the function of TRPV1 and purinergic receptors via Piezo1 activation, thereby mediating chronic pain and itch sensed by the DRG neurons. 

## 2. Materials and Methods

### 2.1. Cell Culture

The human eosinophil cell line AML14.3D10 was cultured in RPMI-1640 Medium (Sigma-Aldrich Co., St. Louis, MO, USA) supplemented with 10% fetal bovine serum (Gibco, Waltham, MA, USA) and 1% penicillin–streptomycin (Gibco, Waltham, MA, USA). Cultures were maintained in a 5% carbon dioxide-enriched atmosphere at 37 °C.

### 2.2. Reagents

Yoda1, Dooku1, Ruthenium Red, and Gadolinium chloride (Gd^3+^) were purchased from Tocris (Bristol, UK) and stock solutions were prepared with dimethyl sulfoxide (DMSO). GsMTx4 were purchased from Abcam (Cambridge, UK). Capsaicin and adenosine 5′-triphosphate disodium salt hydrate (ATP) were purchased from Sigma (St. Louis, MO, USA). Capsaicin stock solutions were prepared with 99.5% ethanol. GsMTx4 and ATP stock solutions were prepared with H_2_O. All stock solutions were stored at −20 °C.

### 2.3. Reverse Transcription-Polymerase Chain Reaction

Total cellular RNA was extracted from cultured eosinophils using TRIzol reagent (Invitrogen, Carlsbad, CA, USA). The first round of polymerase chain reaction (PCR) was performed using 27 μL of PCR buffer containing 10 mM dNTP mix (Bioneer, Daejeon, Republic of Korea), oligo-dT 12–18 primer (Invitrogen, Carlsbad, CA, USA), RNA product, and M-MLV Reverse Transcriptase (200 U/μL) (Invitrogen, Carlsbad, CA, USA). First-strand complementary DNA (cDNA) was utilized for the amplification of glyceraldehyde-3-phosphate dehydrogenase (GAPDH), β-actin, Piezo1, and Piezo2 using PCR. After 35 cycles of amplification, the PCR products were separated on 1.5% agarose gel and stained with eco-dye (BIOFACT, Daejeon, Republic of Korea). The primers used for reverse transcription-PCR (RT-PCR) are listed in Table 1. The protocol included 2 min of initial denaturation at 94 °C, followed by 35 cycles of 45 s of denaturation at 94 °C, 45 s of annealing at 59 °C, and 45 s of elongation at 72 °C, and was completed with 7 min of final elongation.

### 2.4. Immunofluorescence Staining

AML14.3D10 cells attached to Corning™ Cell-Tak Cell and Tissue Adhesive (Corning, NY, USA)-coated coverslips were fixed with 4% paraformaldehyde at 37 °C for 10 min, followed by washing three times with phosphate-buffered saline [PBS (1X); Sigma-Aldrich Co., St. Louis, MO, USA]. The cells were permeabilized with 0.1% Triton X-100 in PBS (1X) for 15 min at room temperature and washed three times with PBS (1X). The cells were blocked with 5% skim milk in PBS (1X) and incubated for 60 min at room temperature. Antibodies were diluted in 5% skim milk in PBS and incubated overnight at 4 °C. The primary antibody was rabbit polyclonal anti-Piezo1 (1:25, Invitrogen, Carlsbad, CA, USA). Cells were then incubated overnight at 4 °C in 5% skim milk in PBS containing the primary antibody. The primary antibody was washed thrice with PBS. The secondary antibody was goat anti-Rabbit IgG (H + L) cross-absorbed secondary antibody Alexa 555 (1:1000, Invitrogen, Carlsbad, CA, USA) applied for 45 min at room temperature and protected from light. The secondary antibody was washed thrice with PBS. The coverslips were air-dried and mounted with Vectashield Hardset Antifade Mounting Medium with DAPI (VECTOR LABORATORIES, Burlingame, CA, USA). Each experiment included a primary omission control where the primary antibody was replaced with only the antibody solution to discern any nonspecific binding of the secondary antibody. Coverslips were mounted over the microscope slides. Analysis was done using a Confocal Laser Scanning Microscope (LSM-710) (Carl Zeiss, Oberkochen, Germany) at the Core Facility for Cell to In Vivo Imaging. 

### 2.5. Supernatant Collection

AML14.3D10 cells were seeded at a density of 2 × 10^5^ cells per well, cultured in fresh medium, and stimulated with Yoda1 or Yoda1 + Dooku1 for 6 h. Subsequently, the cells were centrifuged at 1000 rpm for 3 min to remove the conditional media containing the reagents. Based on research demonstrating the presence of Piezo1 channels in DRG neurons, the supernatant was removed after 6 h of incubation to eliminate Yoda1 particles. The cell pellets were then resuspended in a fresh medium and incubated for an additional 18 h. After incubation, the supernatants were collected for further analysis and stored at −20 °C.

### 2.6. Cell Viability Assay; WST-1 Test

AML14.3D10 cells were incubated in microplates (96-well) in a final volume of 100 μL/well culture medium at 37 °C and 5% CO_2_. Cells were seeded at a concentration of 2 × 10^4^ cells/well in a 100 μL culture medium containing 0.1% DMSO and various concentrations of Yoda1 (final concentration: 5, 10, and 50 μM) into 96-well microplates. The cell culture was incubated for 24 h at 37 °C and 5% CO_2_. After incubation, 10 μL of cell Proliferation Reagent WST-1 (Roche Diagnostics GmbH, Mannheim, Germany) was added and incubated for an additional 2 h at 37 °C and 5% CO_2_. The plates were then shaken thoroughly for 1 min on a shaker. The absorbance of the samples was measured against a background control as a blank using a microplate reader. The wavelength to measure absorbance was 450 nm according to the filters available for the microplate reader with 690 nm used as reference wavelength and subtracted. Results are reported as relative WST-1 activity, where 1.0 corresponds to the absorbance measured in control cultures.

### 2.7. Human Cytokine Array

A human cytokine array (Abcam, Cambridge, MA, USA) was used for the simultaneous detection of 42 cytokines. AML14.3D10 cells were treated with 0.1% DMSO, 10 μM Yoda1 combined with culture medium. Cell lysis was achieved by exposing the cells to the reagents for 24 h. Subsequently, using the bicinchoninic acid method, cell lysate samples were used to determine the total protein concentration. The supernatants were derived by exposing the cells to the reagents for 6 h, and the existing medium was replaced with a fresh culture medium. The cell-cultured media were not diluted. For this assay, cell lysates were diluted to a concentration of 500 ng/mL, while undiluted supernatants were applied at a volume of 1 mL. Both cell lysates and supernatants were added to the prepared cytokine antibody panel membrane, according to the manufacturer’s protocol.

### 2.8. Quantitative Real-Time PCR 

The cDNA synthesized from RNA extracted from AML14.3D10 cells was mixed with SensiFAST™ SYBR Lo-ROX (Meridian Bioscience, Cincinnati, OH, USA), forward and reverse primers, and ultra-pure water (Welgene, Daegu, Republic of Korea) to obtain a total volume of 20 μL in a MicroAmp™ Optical 96-Well Reaction Plate (Invitrogen, Carlsbad, CA, USA). The reactions were performed using a QuantStudio™ 1 Real-Time PCR System (96-well, 0.2 mL; Invitrogen, Carlsbad, CA, USA). The thermal cycling conditions consisted of a hold stage (50 °C for 2 min, 95 °C for 10 min), PCR stage (40 cycles at 95 °C for 15 s, 60 °C for 1 min), and melt curve stage (95 °C for 15 s, 60 °C for 1 min, 95 °C for 1 s). Cycle threshold (Ct) values were obtained, and normalization was performed relative to the Ct value of the housekeeping gene *GAPDH*. The primers used for quantitative real-time PCR (q-PCR) are listed in Table 2.

### 2.9. Mouse DRG Neuron Culture

DRG neurons were harvested from 6–9 week-old mice and incubated with collagenase A (0.2 mg/mL, Roche, Basel, Switzerland)/dispase-II (2.4 units/mL, Roche, Basel, Switzerland) at 37 °C for 90 min. The cells were mechanically dissociated with gentle pipetting. The DRG cells were plated on poly-D-lysine (Sigma-Aldrich Co., St. Louis, MO, USA)-coated coverslips and grown in a neurobasal culture medium with 10% fetal bovine serum (Gibco, Waltham, MA, USA), 2% B27 supplement (Invitrogen, Carlsbad, CA, USA), and 1% penicillin/streptomycin for 1 day. 

### 2.10. Calcium Imaging

#### 2.10.1. AML14.3D10

Live cell calcium imaging was performed using the Olympus BX51WI Fixed Stage Upright Microscope equipped with an IRIS 9 scientific CMOS camera. For calcium ion measurement, cells cultured on coverslips were loaded with a fluorescent calcium indicator, 2 μM Fura-2AM (Invitrogen, Carlsbad, CA, USA), for 1 h. The cell-loaded coverslip was placed in the chamber, and a 2Ca^2+^ buffer {containing 140 mM NaCl, 10 mM 2-[4-(2-hydroxyethyl)piperazin-1-yl]ethanesulfonic acid (HEPES), 10 mM glucose, 2 mM CaCl_2_, 1 mM MgCl_2_, 5 mM KCl; pH = 7.3–7.4, osmolarity = 290–310 mOsmol/kg} was perfused at a rate of 1 mL/min. The emission ratio at wavelengths of 340 nm and 380 nm, indicative of the amount of intracellular calcium ions, was calculated. The ratio of F340/F380 emitted at 510 nm was measured using the CoolLED pE-340 fluorescence measurement device. The ratios were obtained using the MetaFluor program. Yoda1 was applied every 9 min. In the experiment using antagonists, the first application of Yoda1 lasted 30 s, followed by washing the cells with a solution containing 2Ca^2+^. The cells were pre-treated with antagonists for 3 min, followed by immediate application of the solution containing both antagonists and the second application of Yoda1 for 30 s. In the calcium experiment, a third application of Yoda1 was performed for 30 s to confirm recovery of Piezo1 activation.

#### 2.10.2. Mouse DRG Neurons

Wild-type mouse DRG neurons were incubated in the harvested supernatant for 16 h. Live cell calcium imaging was performed using an Olympus BX51WI Fixed Stage Upright Microscope equipped with an intensified camera (optiMOS, QImaging, Surrey, BC, USA). For calcium ion measurement, cells cultured on coverslips were loaded with the fluorescent calcium indicator, 2 μM Fura-2AM (Invitrogen, Carlsbad, CA, USA), for 40 min. The coverslip with the loaded cells was placed in the chamber and a 2Ca^2+^ buffer (containing 140 mM NaCl, 10 mM HEPES, 10 mM glucose, 2 mM CaCl_2_, 1 mM MgCl_2_, 5 mM KCl; pH = 7.3–7.4, osmolarity = 290–310 mOsmol/kg) was perfused at a rate of 1 mL/min. The emission ratio at 340 nm and 380 nm wavelengths, indicative of the amount of intracellular calcium ions, was calculated. The ratio of F340/F380 emitted at 510 nm was measured using a Lambda DG-4 monochromator wavelength changer (Shutter Instrument, Novato, CA, USA). The ratios were calculated using software (Slidebook 6, 3i, Intelligent Imaging Innovations, Denver, CO, USA). An initial application of capsaicin and ATP was performed for 10 s, and the cells were subsequently washed with a solution containing 2Ca^2+^. The neurons were then identified based on their response to a high concentration of KCl, which was used as a neuronal marker.

### 2.11. Statistical Analysis

All data were analyzed using GraphPad Prism 8.0.2 software (GraphPad Software, San Diego, CA, USA). After checking for normality and log-normality using normality and log-normality tests, respectively, parametric or non-parametric tests were utilized accordingly. Parametric and non-parametric tests were performed using the one-way ANOVA for three or more groups. Error bars were visualized using the standard error of measurement (SEM). Significance levels were set as follows: * *p* < 0.05; ** *p* < 0.01; *** *p* < 0.001; **** *p* < 0.0001.

## 3. Results

### 3.1. Physiological Function of Piezo1 in AML14.3D10 Cells

Initially, we examined the RNA and protein levels of Piezo1 using RT-PCR and immunofluorescence to identify the expression of Piezo1 in AML14.3D10 cells. Our results indicated that *PIEZO1*, but not *PIEZO2* mRNA, was expressed in AML14.3D10 cells (Figure 1a). Immunofluorescence images confirmed normal expression of Piezo1 in these cells (Figure 1b). Subsequently, we investigated the physiological role of Piezo1. The Piezo1 agonist, Yoda1, rapidly elevated calcium levels via the Piezo1 channel. Sequential applications of Yoda1 produced a similar amplitude of Ca^2+^ response in the presence of 2 mM extracellular Ca^2+^ in AML14.3D10 cells (Figure 1c,d). Next, we confirmed the degree of inhibition using Piezo1 nonspecific inhibitors. Incubation with another mechanosensitive Ca^2+^ channel blocker, RR (Figure 1e,f), Gd^3+^ (Figure 1g,h), and GsMTx-4 (Figure 1i,j) inhibited the calcium increase induced by Yoda1 (10 μM) by 81%, 77%, and 65%, respectively. Furthermore, we found that Dooku1, an inhibitor of Yoda1-induced Piezo1 activity, significantly reduced the Yoda1-induced Ca^2+^ elevation. This inhibitory effect was reversed to control levels after washout (Figure 1k,l). Dooku1 showed a dose-dependent inhibition pattern, with a half-maximal inhibitory concentration value of approximately 0.5 µM (Figure 1m,n). These results suggest that Piezo1 can be pharmacologically activated to induce Ca^2+^ influx in AML14.3D10 cells.

### 3.2. Identification of Inflammatory Cytokines and Growth Factors Released by AML14.3D10 Cells upon Activation of Piezo1

We performed a cell viability assay to evaluate the cytotoxicity of Yoda1 in these cells. No significant cytotoxicity was observed in cells when treated with 0.1% DMSO, Yoda1 (5–10 μM) for 24 h, although a higher concentration of 50 μM showed noticeable cytotoxicity (Figure 2a). To assess changes in the expression and secretion of cytokines resulting from the activation of the Piezo1 channel, we performed a human cytokine antibody array to screen for cytokine activation (Figure 2b). Quantitative data for 42 targets, including various pro- and anti-inflammatory cytokines and growth factors present in the cells or released into the conditioned media of AML14.3D10 cells with Yoda1, were analyzed (Figure 2c,d). Among the 42 different cytokine antibody spots, the expression levels of 17 pro-inflammatory cytokines, 4 anti-inflammatory cytokines, and 5 growth factors were upregulated or downregulated in cell lysates or supernatants obtained from cells treated with Yoda1, as visualized using heatmaps (Figure 2e,f). These findings indicate that the activation of Piezo1 in AML14.3D10 cells influences the expression and secretion of various cytokine factors.

### 3.3. Identification of Candidate Cytokines in AML14.3D10 Cells upon Activation of Piezo1

Next, we performed real-time PCR using the same samples to validate the results of the cytokine array. qRT-PCR analysis revealed that pro-inflammatory cytokines such as interleukin (IL)-11β, IL-6, IL-8 (C-X-C motif ligand (CXCL) 8), C-C motif ligand (CCL) 5 (RANTES), Caspase-1, and nucleotide-binding domain, leucine-rich–containing family, pyrin domain–containing-3 (NLRP3) (Figure 3a) as well as anti-inflammatory cytokines such as IL-4, IL-10, and transforming growth factor (TGF)-β1 were significantly upregulated in Yoda-treated cells (Figure 3b). This upregulation in gene expression was mitigated in cells treated with Yoda1 and Dooku1. Although significant changes were not observed in cytokines such as IL-12B, Caspase-3, and IL-13, the activation trend of Piezo1 by Yoda1 and Dooku1 was observed. Notably, Figure 3c shows that except for IL-12B, the expression of anti-inflammatory cytokines peaked at 2–4 h and subsequently diminished over time, whereas, except for NLRP3 and CCL5 (RANTES), the expression of pro-inflammatory cytokine increased over time, reaching a peak at 16–24 h. These findings indicate that cytokine expression in AML14.3D10 cells varies over different time points following activation. 

### 3.4. Changes in Capsaicin- and ATP-Induced Calcium Elevation in DRG Neurons Exposed to the Supernatant from Piezo1-Activated AML14.3D10 Cells

To investigate the functional changes related to pain and itch detected by the DRG neurons induced by the activation of AML14.3D10 cells, we followed the protocol illustrated in Figure 4a. We confirmed that the sequential application of capsaicin and ATP induced Ca^2+^ response in the presence of 2 mM extracellular Ca^2+^ in DRG neurons (Figure 4b–e). Using supernatants from non-activated, Yoda1-exposed, or Dooku1/Yoda1-exposed cells, we recorded capsaicin- and ATP-induced calcium influx in DRG neurons after treating them with the respective supernatants for 16 h. We found that DRG neurons treated with supernatants from Yoda1-exposed cells showed a significantly elevated calcium response to capsaicin and ATP compared to the non-activated conditions (Figure 4f–i). These elevations in the calcium level were also inhibited in DRG neurons treated with supernatants from Dooku1/Yoda1-exposed cells. These results suggest that cytokines secreted by AML14.3D10 cells due to Piezo1 activation enhance the activity of TRPV1 and purinergic receptors in DRG neurons. 

## 4. Discussion

This study is the first to elucidate the functional role of Piezo1 in both the immune and sensory neuronal pathways. Specifically, we observed that Yoda1 induced calcium influx via Piezo1 activation in AML14.3D10 eosinophil cell lines, playing a crucial role in the pathological functions related to cytokine release. Activated AML14.3D10 cells release various cytokines into the surrounding medium, which enhance the activation of TRPV1 and ATP receptors, both of which are associated with pain or itch [24,25,27] detected by the DRG neurons. These findings demonstrate that inflammatory substances released from activated eosinophils modulate the activation of TRPV1 and ATP receptors in DRG neurons. Thus, the intimate interaction between eosinophil cells and DRG neurons can be elucidated through the regulation of pathological conditions mediated by Piezo1. 

Immune cells critically depend on receiving mechanical inputs from the extracellular environment, including stretch, shear force, elasticity, and matrix stiffness [29]. Mechanical forces can influence the functions of different immune cell types [30]. Mechanical channels have been identified in a wide range of immune cells, not limited to macrophages and neutrophils, but also in monocytes and eosinophils [10,11,30,31]. However, the characteristics of these mechanical channels in immune cells have not been fully elucidated. Macrophages can differentiate into the pro-inflammatory (M1 type) or pro-healing (M2 type) phenotype in response to stiffness conditions [32]. Neutrophils also secrete cytokines in response to mechanical deformation [31]. In addition, this activation of mechanosensitive ion channels integrates with that of pro-inflammatory mediators to modulate macrophage inflammatory responses (M1/M2 phenotypes) [29]. Since macrophages and neutrophils can respond to multiple signals and polarize into different phenotypes, these cells influence both physiological and pathological responses [23,29,33]. Thus, mechanosensitive mechanisms regulate myriad cellular functions involved in cytokine secretion [10]. Mechanosensitive ion channels, such as TRPV4, which controls calcium influx, play a role in mechanotransduction in immune cells [34]. Recent studies have recognized the important roles of Piezo1 as a mechanosensor of stiffness and its modulatory role in the polarization response of macrophages [33]. These findings underscore the importance of focusing on their polarization following Piezo channel activation. 

Piezo channels are thought to play roles in various diseases and are triggered by mechanical changes in tissue [15]. Recent studies have reported that Piezo channels that are expressed in many types of immune cells can significantly affect the functionality of these cells [10]. Specifically, Piezo1 and 2 are mechanically gated ion channels activated directly by mechanical stretch and chemical stimuli [29]. Piezo1 is implicated in blood pressure regulation and myoblast fusion during skeletal muscle formation [35], while Piezo2 is involved in the regulation of light touch sensation and proprioception [36,37]. Notably, Piezo1 activation is directly induced by mechanical stretch of the lipid bilayer, facilitating calcium flow across the membrane [38]. In the absence of mechanical stimuli, Piezo1 can be activated by small molecules such as Yoda1 [39]. Although the possibility of Piezo2 expression exists, we initially confirmed the absence of Piezo2 expression through RT-PCR analysis. Furthermore, we found that the non-selective antagonists—Ruthenium Red (10 μM), Gadolinium chloride (Gd^3+^) (10 μM), and GsMTx4 (1 μM)—effectively inhibited the calcium increase induced by Yoda1 (10 μM). Notably, Dooku1, an inhibitor of Yoda1-induced Piezo1 activity, significantly reduced the Ca^2+^ elevation caused by Yoda1. Therefore, our experimental results indicate that the physiological functions of Yoda1 are specifically attributable to Piezo1, and even if Piezo2 is expressed, its influence is likely to be minimal. 

Eosinophils can synthesize, store within intracellular granules, and secrete highly diverse types of cytokines in both health and disease [40]. Eosinophils were thought to counteract inflammation in inflamed tissue, potentially serving at the root of the resolution of altered airway function [41]. Eosinophils secrete IL-10 after stimulation with IL-4 or IL-12, and IL-10 is a potent anti-inflammatory cytokine that prevents autoimmune and allergic inflammatory responses by downregulating the functions of monocytes, macrophages, and dendritic cells [41,42,43]. However, eosinophils are also thought to play a detrimental role as terminal effects or cells, because activated eosinophils can secrete a wide array of proinflammatory cytokines (including IL-2, IL-4, IL-5, IL-10, IL-12, IL-13, IL-16, IL-18, and TGF-α/β), chemokines (CCL5(RANTES) and eotaxin-1), and lipid mediators (platelet-activating factor and leukotriene C4) [8,41]. Numerous studies have elucidated that eosinophils comprise specific granule contents and express a diverse array of receptors, which are hallmarks of these cells [44,45]. Upon activation, eosinophils release potent mediators, including cytokines that can directly interact with and modulate corresponding receptors on pain or itch-mediating DRG neurons [8,40,46,47]. In our experiments with AML14.3D10 cells, we observed changes in the expression of 26 proteins related to pro-inflammatory, anti-inflammatory, and growth factors following Piezo1 activation. Subsequently, we investigated the modulation of gene expression for various cytokines using the activator and inhibitor of Piezo1, Yoda1, and Dooku1, respectively. While the changes in gene and protein expression did not exhibit a consistent pattern, it appears that the regulation of cytokine expression in AML14.3D10 cells is influenced by the level of Piezo1 activity. 

The DRG neurons constitute a major site of nociceptive processing under pathological conditions [48]. Various proteins such as the TRP channel, purinergic receptor, voltage-gated ion channels, glutamate receptor, GABA receptor, Piezo1 receptor, and G-protein-coupled receptors can induce chronic conditions such as pain and itch in the peripheral nervous system [48,49,50,51,52,53]. The DRG serves as a critical site for processing nociception, particularly in pathological conditions. Various proteins, including the TRP channel, purinergic receptor, voltage-gated ion channels, glutamate receptor, GABA receptor, Piezo1 receptor, and G-protein-coupled receptors, can induce chronic pain and itch in the peripheral nervous system. Hence, we examined the modulation of key receptors in the DRG exposed to supernatants from Piezo1-activated AML14.3D10 cells. TRPV1 is mainly distributed in small- and medium-diameter DRG and plays a crucial role in enhancing pain sensitivity induced by heat and chemicals, with primary sensory neurons being pivotal in the mechanisms of pain generation and conduction [54,55]. Extracellular ATP activates downstream signaling by engaging ionotropic P2X and metabotropic P2Y receptors on cell membranes, which can evoke pain sensation; seven mammalian P2X receptor subtypes (P2X1–P2X7) are mainly expressed in DRG neurons [56]. P2X receptors are abundant in DRG neurons, and P2X3 sensitization reportedly causes inflammatory and neuropathic pain [57]. P2Y receptors, classified by sequence and G protein selectivity, either activate phospholipase C (P2Y1, P2Y2, P2Y4, P2Y6, P2Y11) or inhibit adenylate cyclase (P2Y12, P2Y13, P2Y14). 2Y12 is linked to chronic itching in type 2 diabetes, as indicated by increased DRG expression in diabetic rats [58,59]. These results suggest that the heightened activity of TRPV1 and the purinergic receptor in DRG neurons near Piezo1 activation of AML14.3D10 cells may have an impact on neuropathological conditions.

Recent studies have demonstrated that the activation of immune cells under various peripheral conditions significantly influences DRG neurons, contributing to the modulation of pain and sensory responses [60,61,62]. Immune cell-derived mediators can interact with DRG neurons, altering their excitability and function [47,63]. Our experiment specifically targeted certain cytokines and growth factors among the numerous substances secreted by eosinophils. Among the substances that were not detected, notably neurotrophins, such as nerve growth factor (NGF) and neurotrophin-3 (NT-3), have been reported to be released by eosinophils and to enhance allergic disease responses through neurologic mechanisms [64]. Therefore, further investigation into these candidate substances that may influence neuronal cells is crucial. To investigate this interaction, in our experiment, DRG neurons were treated with supernatant derived from activated eosinophils through the physiological function of Piezo1, followed by testing the responses of TRPV1 and ATP receptors. Our results showed a significant increase in TRPV1 and ATP receptor responses owing to cytokine changes in the Yoda1-treated eosinophils. This indicates that the substances released by eosinophils upon Piezo1 activation modulate DRG neuron activity, suggesting a possible mechanism for immune cell influence on sensory neurons, which contribute to pain and sensory disorders. The differential cytokine secretion by eosinophils via Piezo1 activation highlights the interplay between immune cells and sensory neurons. Given the differences in complexity of in vitro and in vivo environments, further research is needed to clarify specific pathways and therapeutic targets in pathological conditions.

## 5. Conclusions

Our study elucidates the functional role of Piezo1 in the human eosinophil cell line AML14.3D10. We demonstrate that Piezo1 is expressed in these cells and that its activation by Yoda1 induces significant calcium influx. Activation of Piezo1 modulates the expression and secretion of various cytokines. Notably, supernatants from Piezo1-activated AML14.3D10 cells enhance capsaicin and ATP-induced calcium responses in DRG neurons, emphasizing Piezo1’s role in immune cells and its impact on sensory neurons.

## Figures and Tables

**Figure 1 biomolecules-14-01157-f001:**
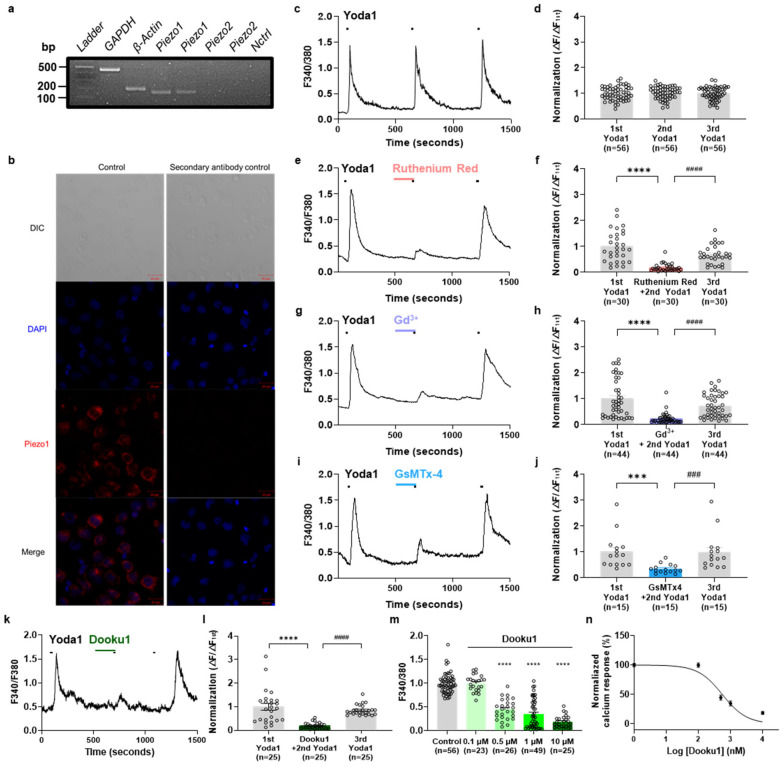
AML14.3D10 cells exhibit functional expression of the Piezo1 channel. (**a**) Reverse transcription-polymerase chain reaction (RT-PCR) analysis of the AML14.3D10 cells shows the expression of transcripts using two different primer pairs specific for *Piezo1* and *Piezo2*. (**b**) Representative immunofluorescence (IF) images of AML14.3D10 cells with primary and secondary antibodies showing Alexa 555 with the indicated markers (scale bar, 20 μm). Negative controls represent AML14.3D10 cells exposed to Alexa 555 secondary antibody only (no primary antibody) in addition to DAPI. (**c**) Calcium imaging showing Ca^2+^ response induced by sequential application of Yoda1 (10 μM, 30 s). (**d**) Mean normalized amplitude of sequential Yoda1-induced Ca^2+^ response (*n* = 56). (**e**) Representative traces showing Yoda1- (10 μM, 30 s) and Ruthenium Red-induced (10 μM, 3 min pre-treatment, Pink) Ca^2+^ response. (**f**) Mean normalized amplitude of Ca^2+^ response induced by sequential application of Yoda1 (10 μM, 30 s) and Ruthenium Red (10 μM, 3 min pre-treatment). (**g**) Representative traces showing Yoda1- (10 μM, 30 s) and Gd^3+^ (10 μM, 3 min pre-treatment, light purple) Ca^2+^ response. (**h**) Mean normalized amplitude of Ca^2+^ response induced by sequential application of Yoda1 (10 μM, 30 s) and Gd^3+^ (10 μM, 3 min pre-treatment). (**i**) Representative traces showing Yoda1- (10 μM, 30 s) and GsMTx4-induced (1 μM, 3 min pre-treatment, blue) Ca^2+^ response. (**j**) Mean normalized amplitude of Ca^2+^ response induced by sequential application of Yoda1 (10 μM, 30 s) and GsMT × 4 (1 μM, 3 min pre-treatment). (**k**) Representative traces showing Yoda1- (10 μM, 30 s) and Dooku1-induced (10 μM, 30 s, Green) Ca^2+^ response. (**l**) Mean normalized amplitude of Ca^2+^ response induced by sequential application of Yoda1 (10 μM, 30 s) and Dooku1 (10 μM, 3 min pre-treatment). (**m**) Mean normalized amplitude of sequential Yoda1- and Dooku1-induced Ca^2+^ response in the control (n = 56), Dooku1 0.1 μM (*n* = 23), 0.5 μM (*n* = 26), 1 μM (*n* = 49), and 10 μM (*n* = 25). One-way ANOVA (Dunn’s multiple comparisons test, **** *p* < 0.001). (**n**) Graph showing the Ca^2+^ response according to the concentration of Dooku1 on the IC_50_ curve. All results are presented as the mean ± standard error of the mean (SEM). *** *p* < 0.001, **** *p* < 0.0001, Dunnett’s test following one-way analysis of variance versus the first Yoda1 treated group; ^###^
*p* < 0.001, ^####^
*p* < 0.0001, versus the mechanosensitive Ca^2+^ channel blocker (Ruthenium Red, Gd^3+^, GsMTx4) and Dooku1 group. IC_50_: half-maximal inhibitory concentration, Gd^3+^: Gadolinium chloride. Original images of (**a**) can be found in Appendix A.

**Figure 2 biomolecules-14-01157-f002:**
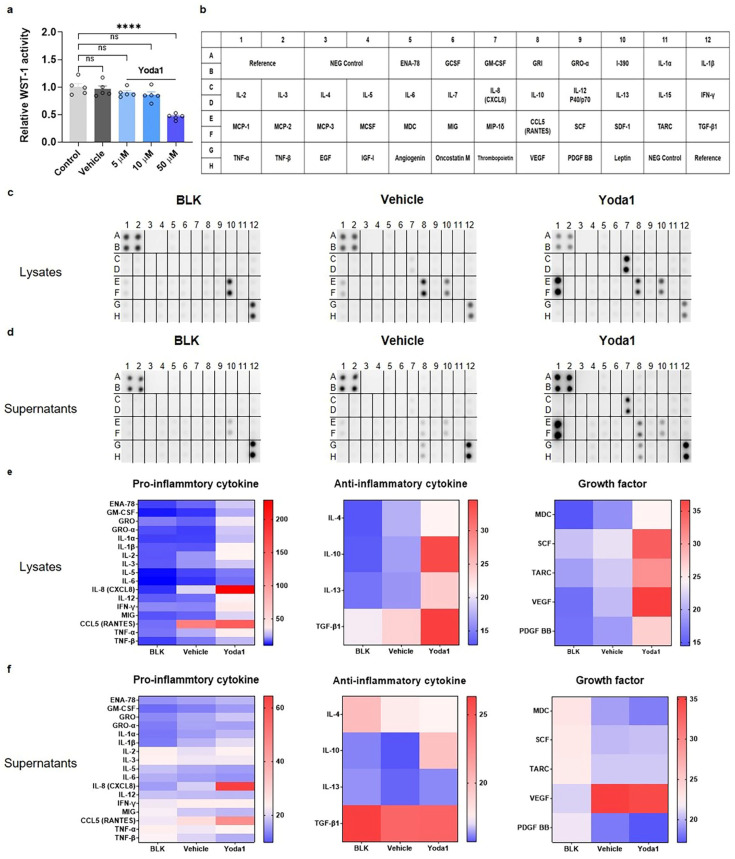
Piezo1 activation within AML14.3D10 cells induces various changes in cytokine expression and secretion. (**a**) AML14.3D10 cells were seeded at a density of 2 × 10^4^ cells per well in 100 μL of culture medium in 96-well transparent plates with 0.1% DMSO, Yoda1 5, 10, and 50 μM for 24 h. The WST-1 activity of 0.1% DMSO and Yoda1 at 5 μM, 10 μM, and 50 μM, normalized to the WST-1 activity of the control, is 0.957, 0.897, 0.857, and 0.459, respectively. Results are presented as the mean ± standard error of the mean (SEM). Dunnett’s test following one-way analysis of variance **** *p* < 0.001 compared with control group (n = 5). (**b**) Reference map for cytokine array, adapted from the manufacturer’s information. Complete array images from (**c**) lysates and (**d**) supernatants of eosinophils. The human cytokine array of lysate proteins from eosinophils cultured in 0.1% DMSO and Yoda1 media. The human cytokine array of supernatants from eosinophils cultured in 0.1% DMSO and Yoda1 media. BLANK (BLK) represents an uncultured RPMI1640 medium. (**e**) Heat map showing the relative cytokine concentration with selected inflammatory cytokines, anti-inflammatory cytokines, and growth factors in lysate proteins from stimulated eosinophils after 24 h. (**f**) Heat map showing the relative concentration of cytokines with selected inflammatory cytokines, anti-inflammatory cytokines, and growth factors in supernatants from stimulated eosinophils after 24 h. All results are presented as the mean ± standard error of the mean (SEM). **** *p* < 0.0001, ns, Dunnett’s test following one-way analysis of variance versus the control group. Ns indicates no significant difference. DMSO: dimethyl sulfoxide.

**Figure 3 biomolecules-14-01157-f003:**
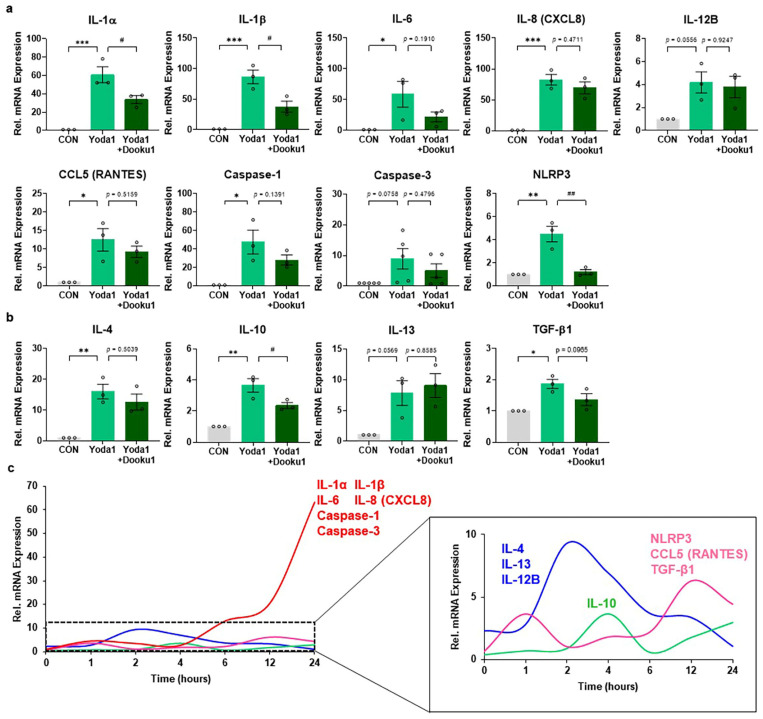
Piezo1 activation within AML14.3D10 cells induces various changes in cytokine expression over different time intervals. Quantitative real-time polymerase chain reaction analysis was performed for the following. (**a**) Pro-inflammatory cytokines: IL-1α (24 h, *n* = 3), IL-1β (24 h, *n* = 3), IL-6 (24 h, *n* = 3), IL-8 (CXCL8) (24 h, *n* = 3), IL-12B (2 h, *n* = 3), CCL5 (RANTES) (12 h, *n* = 3); Inflammasome: Caspase-1 (24 h, *n* = 3), Caspase-3 (24 h, *n* = 5), and NLRP3 (12 h, *n* = 3). (**b**) Anti-inflammatory cytokines: IL-4 (2 h, *n* = 3), IL-10 (4 h, *n* = 3), IL-13 (2 h, *n* = 3), TGF-β1 (12 h, *n* = 3). (**c**) Time course of gene expression induced by Piezo1 activation in AML14.3D10 cells. Each time point value is an average of data: 2-h time point (Blue, IL-4, IL-13, IL-12B), 4-h time point (Green, IL-10), 12-h time point (Pink, NLRP3, CCL5 (RANTES), TGF-β1), 24-h time point (Red, IL-1α, IL-1β, IL-6, IL-8 (CXCL8), Caspase-1, Caspase-3). Data are presented as the mean ± standard error of the mean. * *p* < 0.05, ** *p* < 0.01, *** *p* < 0.001, Dunnett’s test following one-way analysis of variance versus the control group; ^#^
*p* < 0.05, ^##^
*p* < 0.01, versus the Yoda1 group. CXCL: C-X-C motif ligand, CCL: C-C motif ligand, NLRP3: nucleotide-binding domain, leucine-rich–containing family, pyrin domain–containing-3, TGF: transforming growth factor, IL: interleukin.

**Figure 4 biomolecules-14-01157-f004:**
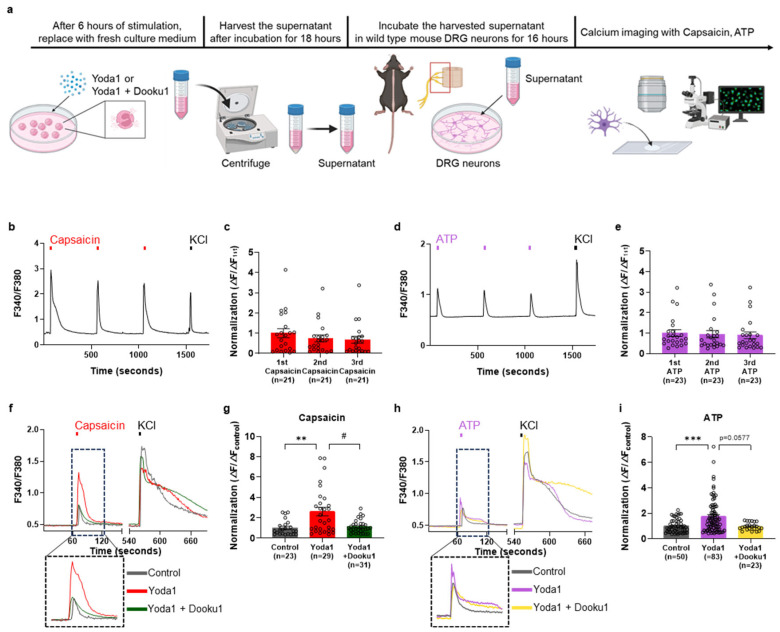
Secreted supernatant induces capsaicin- and ATP-induced calcium increase in DRG neurons. (**a**) Schematic illustration of the experimental design used in this study. (**b**) Representative traces induced by capsaicin (100 nM, 10 s) and potassium chloride (KCl, 50 mM, 10 s) in mouse DRG neurons. (**c**) Mean normalized amplitude of the capsaicin-induced Ca^2+^ response (Red, *n* = 21). (**d**) Representative traces of responses to ATP (100 μM) and potassium chloride (50 mM) in mouse DRG neurons. (**e**) Mean normalized amplitude of the ATP-induced Ca^2+^ response (Purple, *n* = 23). (**f**) Representative traces of capsaicin-evoked calcium responses in mouse DRG neurons with control (Dark gray, *n* = 23), Yoda1-stimulated supernatant (Red, *n* = 29), and Yoda1 + Dooku1-stimulated supernatant (Green, *n* = 31). (**g**) Mean normalized amplitude of capsaicin-induced Ca^2+^ response. (**h**) Representative traces of ATP-evoked calcium response in mouse DRG neurons with control (Dark gray, *n* = 50), Yoda1-stimulated supernatant (Purple, *n* = 83), Yoda1 + Dooku1-stimulated supernatant (Yellow, *n* = 23). (**i**) Mean normalized amplitude of ATP-induced Ca^2+^ response. Electrically excitable neurons respond to depolarization by 50 mM potassium chloride. All results are presented as the mean ± standard error of the mean (SEM). ** *p* < 0.01, *** *p* < 0.001, Dunnett’s test following one-way analysis of variance versus the control group; ^#^
*p* < 0.05, versus the Yoda1 group. DRG: dorsal root ganglia, ATP: adenosine 5′-triphosphate disodium salt hydrate.

**Table 1 biomolecules-14-01157-t001:** Reverse transcription-polymerase chain reaction primer list.

Target Gene (Product Length)		Forward (5′-3′)	Reverse (5′-3′)
*Human GAPDH*	459 bp	CAA ATT CCA TGG CAC CGT CA	ATG ATG TTC TGG AGA GCC CC
*Human* *β-Actin*	197 bp	GCC GAC AGG ATG CAG AAG GAG ATC A	AAG CAT TTG CGG TGG ACG ATG GA
*Human* *PIEZO1*	165 bp	ACT TTC CCA TCA GCA CTC GG	CCA CGA AGT CCT TGA GAC CC
*Human* *PIEZO1*	168 bp	TTC CCC AAC AGC ACC AAC TT	CAC GAT GGC CTC GAA TAC CA
*Human* *PIEZO2*	165 bp	ATG GCC TCA GAA GTG GTG TG	ATG TGG TTG CAT CGT CGT TTT
*Human* *PIEZO2*	176 bp	CAT AGT GAA CCC GGA CCT GT	CCG CTG TTA TTT GGA TGG GG

**Table 2 biomolecules-14-01157-t002:** Quantitative real-time polymerase chain reaction primer list.

Target Gene (Product Length)		Forward (5′-3′)	Reverse (5′-3′)
*Human GAPDH*	171 bp	GGA TTT GGT CGT ATT GGG CG	CTT CCC GTT CTC AGC CTT GA
*Human IL-1* *α*	147 bp	AGA TGC CTG AGA TAC CCA AAA CC	CCA AGC ACA CCC AGT AGT CT
*Human IL-1* *β*	172 bp	CCA AAC CTC TTC GAG GCA CA	GGG CCA TCA GCT TCA AAG AAC
*Human IL-6*	110 bp	ATG CAA TAA CCA CCC CTG AC	AAA GCT GCG CAG AAT GAG AT
*Human IL-8* *(Human CXCL8)*	134 bp	GTG CAG TTT TGC CAA GGA GT	AAT TTC TGT GTT GGC GCA GT
*Human IL-12b*	159 bp	TCA CAA AGG AGG CGA GGT TC	CAG CAG GTG AAA CGT CCA GA
*Human Caspase-1*	132 bp	TGG ATA AGA CCC GAG CTT TG	CCT GAG GAG CTG CTG AGA GT
*Human Caspase-3*	152 bp	AAA ATA CCA GTG GAG GCC GA	ATT CTG TTG CCA CCT TTC GG
*Human CCL5* *(Human RANTES)*	166 bp	CTG CTT TGC CTA CAT TGC CC	CTT GTT CAG CCG GGA GTC AT
*Human NLRP3*	144 bp	CCG ACC TCA GGA ATC ATG GA	GGT AGT ACA TGG CGG CAA AG
*Human IL-4*	152 bp	CTT TGC TGC CTC CAA GAA CA	TCC TGT CGA GCC GTT TCA G
*Human IL-10*	156 bp	TGA TGC CCC AAG CTG AGA AC	AGG CAT TCT TCA CCT GCT CC
*Human IL-13*	150 bp	CAG AGG ATG CTG AGC GGA TT	GTT GAA CTG TCC CTC GCG AAA
*Human TGF-* *β* *1*	152 bp	ACT CGC CAG AGT GGT TAT CT	GGT AGT GAA CCC GTT GAT GT

## Data Availability

The original data accumulated during the study are included in the article, and further inquiries can be directed to the corresponding authors.

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
