# Peer review of "Functional Role of Piezo1 in the Human Eosinophil Cell Line AML14.3D10: Implications for the Immune and Sensory Nervous Systems"

_biomolecules, 2024, doi:10.3390/biom14091157_

Round 1
Reviewer 1 Report
Comments and Suggestions for Authors
The study reports the novel finding that Piezo1 is expressed and functional in an eosinophil cell line. Piezo1 engagement by a selective agonist results in the secretion of a series of proinflammatory mediators from a eosinophil cell line (AML14.3D10) that may sensitize responses to TRPV1 and ATP receptor in DRG neurons, underlying the possible role of Piezo1-derived mediators in eosinophils in increasing pain or itch responses.
The paper is of interest and open new avenues for future investigation. However, there some points that deserve consideration.
1. While data supporting the expression/function of Piezo1 in the AML14.3D10 cells is convincing, some key experiments of Piezo1 expression/function in eosinophils may markedly strengthen the message of the paper;
2. Many cytokines/growth factors may sensitize DRG neurons, and in particular increase proalgesic responses mediated by TRPV1. Among them neurotrophins are of paramount importance. NGF can be released from eosinophils (NGF and NT-3, DOI:10.1182/blood.v99.6.2214). Thus, investigation on NGF and related molecules may complete the panel of proinflammatory/proalgesic mediators released from AML14.3D10 upon Piezo1 activation;
3. The sentence ‘Although we did not detect modulation 440 of TRPA1, PIEZO1, GABA, or glutamate receptor activity,’ in the Discussion, is not supported by experimental data as I did not find any mention of related experiments in the Results section.
4. In the limitation of the study it should be mention the failure of the Piaezo2 antagonist to reduce increases induced by Yoda1 in many instances. This casts doubts on a genuine Piezo1 mediated response. A comment on this should be added to the Discussion.
Comments on the Quality of English LanguageThe English text can be ameliorated. For example 'to play a detrimental 415 role as terminal effects cells, because should be to play a detrimental 415 role as terminal effectsorcells, because
Reviewer 2 Report
Comments and Suggestions for Authors
Reviewer Comments to Authors:
The manuscript by Sung-Min Hwang and Ji-Min Song et al. is devoted to studying influence of Piezo1 in cytokine secretion in eosinophils on signaling pathways with TRPV4 and purinergic receptors and role of Piezo1 in physiological processes. The presented research is written well and structured, about 50% of cited references are mostly recent. In reviewer’s opinion the manuscript can be published in the ‘Biomolecules’ after clarification and correction of the text.
Please find the comments outlined below.
General comments:
1. Explain the choice of concentration of Piezo1 antagonists (Ruthenium Red, Gd3+, GsMTx-4) used in Ca-measurements. The applying concentrations are in 25-30 times less than concentrations that completely block the Piezo1 activity.
2. The duration of Ca-measurements is more than 25 minutes. How the authors control essential parameters such as temperature and level of CO2.
3. The authors provide representative Ca2+-responses. Based on the graphs, we can see various responses to the application of selective agonist Yoda1. Is it possible to assume that the cell line is heterogeneous in Piezo1 channels?
4. The bar graph on the Figure 1m shows a wide range of values for Dooku1 (1 uM), how is the data distributed?
5. How much DMSO did the control contain? Possibly the decrease in cell viability is due to the toxic effects of DMSO (Figure 2A).
6. Could the authors explain in more detail why the time periods of 6 hours and 18 hours were chosen when collecting the supernatant.
Specific comments:
1. Provide the important information about reverse transcription PCR such as reaction conditions, temperature and time.
2. Check the similar spelling of Piezo1 channel and Yoda1 through the manuscript.
3. In Ca-measurements (Figure 1) set the same range of scale bars, it is difficult to compare responses.
4. Indicate in the text and on the graph the Figure 3e the same name of the chemokine IL-8 (or IL-8 or CXCL8) and CCL5 (RANTES).
